# Prevalence of non-communicable diseases risk factors and their determinants: Results from STEPS survey 2019, Nepal

**Bihungum Bista**[1], **Meghnath Dhimal**[1]*, **Saroj Bhattarai**[1], **Tamanna Neupane**[1], **Yvonne Yiru Xu**[2], **Achyut Raj Pandey**[1], **Nick Townsend**[3], **Pradip Gyanwali**[1], **Anjani Kumar Jha**[1]

**1** Nepal Health Research Council, Ramshah Path, Kathmandu, Nepal, **2** WHO South East Asia Regional Office, New Delhi, India, **3** Department for Health, University of Bath, Bath, United Kingdom

☯ These authors contributed equally to this work.

* meghdhimal@gmail.com

**Data Availability Statement:** All relevant data are within the manuscript and its Supporting information files.

## Abstract

### Background

The World Health Organization (WHO) recommends ongoing surveillance of non-communicable diseases (NCDs) and their risk factors, using the WHO STEPwise approach to surveillance (STEPS). The aim of this study was to assess the distribution and determinants of NCD risk factors in Nepal, a low-income country, in which two-thirds (66%) of annual deaths are attributable to NCDs.

### Methods

A nationally representative NCD risk factors STEPS survey (instrument version 3.2), was conducted between February and May 2019, among 6,475 eligible participants of age 15–69 years sampled from all 7 provinces through multistage sampling process. Data collection involved assessment of behavioral and biochemical risk factors. Complex survey analysis was completed in STATA 15, along with Poisson regression modelling to examine associations between covariates and risk factor prevalence.

### Results

The most prevalent risk factor was consumption of less than five servings of fruit and vegetables a day (97%; 95% CI: 94.3–98.0). Out of total participants, 17% (95% CI: 15.1–19.1) were current smoker, 6.8% (95% CI: 5.3–8.2) were consuming ≥60g/month alcohol per month and 7.4% (95% CI:5.7–10.1) were having low level of physical activity. Approximately, 24.3% (95% CI: 21.6–27.2) were overweight or obese (BMI≥25kg/m$^2$) while 24.5% (95% CI: 22.4–26.7) and 5.8% (95% CI: 4.3–7.3) had raised blood pressure (BP) and raised blood glucose respectively. Similarly, the prevalence of raised total cholesterol was 11% (95% CI: 9.6–12.6). Sex and education level of participants were statistically associated with smoking, harmful alcohol use and raised BP. Participants of age 30–44 years and 45–69 years were found to have increased risk of overweight, raised BP, raised blood sugar

**Funding:** This survey was funded by the Government of Nepal and the World Health Organization.

**Competing interests:** The authors have declared that no competing interests exist.

and raised blood cholesterol. Similarly, participants in richest wealth quintile had higher odds of insufficient physical inactivity, overweight and raised blood cholesterol. On average, each participant had 2 NCD related risk factors (2.04, 95% CI: 2.02–2.08).

## Conclusion

A large portion of the Nepalese population are living with a variety of NCD risk factors. These surveillance data should be used to support and monitor province specific NCD prevention and control interventions throughout Nepal, supported by a multi-sectoral national coordination mechanism.

## Introduction

Non-communicable diseases (NCDs) are the leading causes of disease burden worldwide [1]. According to World Health Organization (WHO) estimates, NCDs are responsible for 71% of all deaths globally, with around 85% of premature deaths from NCDs occurring in low and middle income countries (LMICs) [2]. Behavioral risk factors including smoking, alcohol consumption, unhealthy diet and physical inactivity, along with biological risk factors such as raised blood pressure (BP), blood glucose and cholesterol level, along with overweight and obesity have been identified as the major underlying causes of NCDs [3]. In addition, the risk of progression of NCDs is reported to increase with the co-existence of multiple risk factors within the same individual, which is referred to as clustering [4–6].

Data from Nepal, a lower middle income country in South Asia, indicate an 8% increase in deaths caused by NCDs between 2014 and 2016 [7,8] with two-thirds (66%) of the 182,751 deaths recorded in Nepal in 2017, attributed to NCDs [1]. A 2019 population based nationwide cross-sectional study in Nepal also indicating the high burden of NCDs with a high prevalence of COPD, diabetes, chronic kidney disease, and coronary artery disease, which could pose a serious challenge to health systems in the near future. Apart from these diseases, diabetes mellitus is recognized to affect a notable proportion (8.5%) of the adult population in Nepal [9]. The 2013 STEPS Survey in Nepal also confirming the high prevalence of various risk factors including smoking (19%), low consumption of fruits and vegetables (99%), raised BP (26%), and abnormal lipids (23%) [10]. Likewise, a substantial proportion of the Nepalese population was found to be hypertensive (19.9%) with more than one fifth overweight or obese (21.4%) [11].

To combat NCDs at a population level, the Nepal government adopted a Multisectoral Action Plan for the Prevention and Control of Non-Communicable Diseases in 2014 [12], aligning with the NCD global monitoring framework [13]. One of the key activities identified and included in the multisectoral action plan was to have a periodic NCD STEPS survey to track progress on prevention and control of NCDs within the country. With recent transition to federal structure, Nepal also needs evidence on NCD risk factors at provincial level so as to facilitate decision making process in health sector. In this context, this study aimed to assess the epidemiological distribution and determinants of behavioral (tobacco, alcohol, diet, salt consumption, physical activity) and biological risk factors (overweight/obesity, raised BP, raised blood sugar and cholesterol levels) associated with major/selected NCDs in.

## Methods

### Study settings

Nepal is a landlocked country situated in Southern Asia between India and China. The country runs from a plain area in the south, known as Terai, to the mountainous area of the Himalayas in the north, with a hilly region in between the two. Administratively, Nepal is comprised of 7 provinces, 77 districts and 753 local bodies.

### Study design and sampling techniques

It was a nationally-representative cross-sectional NCD risk factors survey, following the WHO STEPwise approach to surveillance (STEPS), an integrated surveillance tool through which countries can collect, analyse and disseminate core standardized information on NCDs [14]. Data for the survey was collected from the eligible adult population, aged between 15 and 69 years, between February and May 2019.

Sampling for the survey took into consideration the current federal structure of Nepal, such that findings could be generalized to the provincial levels. A multistage cluster sampling method was used to select 6,475 eligible participants across all 7 provinces in Nepal. A total of 259 wards were selected as the primary sampling units (PSU) at the first stage, maintaining 37 PSUs from each province. The household listing operation was carried out in 259 PSUs, in order to develop a sampling frame for selection of individual households at the second stage. From the prepared list of the households, 25 households per PSU were sampled using systematic random sampling, after determining the sampling interval by dividing the number of listed households by 25. From each of the selected households, one adult member of age 15–69 years was sampled randomly for participation in the survey using an android tablet. This household listing process provided greater rigor to the sampling process than for previous STEPS surveys. Further details on the sampling process can be found elsewhere [14].

**Table 1. Variables definition.**

| Variables | Definitions |
|---|---|
| Current smoker | Participants those who had smoked in the past 30 days were considered as current smoker for this survey. |
| Harmful use of alcohol | Consumption of $\geq$60 gm of pure alcohol on an average day in the past 30 days was considered harmful use. |
| Insufficient fruits and vegetables intake | Participants who ate less than five servings of fruits and vegetables per day were considered to have insufficient fruit and vegetable intake. |
| Insufficient Physical activity | Participants who participated in less than the equivalent of 150 minutes of moderate intensity (600 METs) physical activity per week were categorized as having insufficient physical activity. |
| Overweight | Participants with a BMI $\geq$ 25 kg/m$^2$, had classified them as being overweight. |
| Raised BP | Participants were classified as having raised BP if the average 2nd and 3rd measurement of systolic BP was $\geq$140 mmHg, or the average diastolic BP was $\geq$90 mmHg, or if they reported to be taking antihypertensive medication. |
| Raised blood sugar | Participants with a fasting blood sugar $\geq$126 mg/dl, or those currently taking medications to lower blood sugar, were considered to have raised blood sugar. |
| Raised blood cholesterol | Participants whose blood cholesterol was above 190 mg/dl, or those currently taking medications to lower blood cholesterol, were considered to have raised blood cholesterol |

## Variable definition

For this study, current smoking, harmful use of alcohol, insufficient fruit and vegetable intake, insufficient physical activity are considered as a behavioral factor. Similarly, overweight and raised BP, are categorized as a physical factor. Raised blood sugar and raised blood cholesterol together are considered as biochemical factor. The operational definitions of the outcome variables (NCD risk factor) are presented in Table 1.

## Data collection

We conducted face to face interviews using standardized questions from the STEPS Survey (version 3.2) [15]–an update on the 2013 STEPs survey. The survey collected information related to behavioral (tobacco use, alcohol use, physical activity, fruits and vegetables intake) (STEP I), physical (height, weight and BP) (STEP II) and biochemical measures (Blood sugar, sodium level measurement in urine) (STEP III). Measurement of height, weight (measured using SECA weighing machine, Germany), BP (measured using OMRON BP monitor), blood sugar (measured using Cardiocheck PA) and blood cholesterol (measured using Cardiocheck PA) were made as per the WHO STEPS manual. Details of the measurement process has been described in more detail elsewhere [14,16].

The survey also included questions related to tobacco policy, alcohol policy and programs. Furthermore, it included questions related to violence and injury, along with musculoskeletal pain. In addition, in this round of the STEPs survey dietary salt intake level was estimated via spot urine collection, along with that concentrations of blood glucose and total cholesterol was measured using CardioCheck, PA, as recommended by the WHO.

## Statistical analysis

Analysis was performed with STATA version 15.1 using survey (*svy*) set command, defining clusters and sampling weight information. All estimates were weighted by sample weights and are presented with 95% confidence intervals (CI). Prevalence estimates were calculated using Taylor series linearization. Chi-square tests were used for bivariate analysis, to test associations between independent and dependent variables. Furthermore, Poisson regression was used to calculate the adjusted prevalence ratio (APR) between each NCD risk factors and sociodemographic covariates (age, sex, education, marital status, province, ecological belt and place of residence) included simultaneously [17]. For clustering analysis of NCD risk factors, the numbers of risk factors present within each participant were summed (from 0 to 5) and was analyzed against socio-demographic covariates through Poisson regression. The relationship between the number of risk factors and covariates was estimated through adjusted relative risk ratios (ARR), with the number of risk factors designated as the dependent variable.

## Ethical considerations

Ethical approval to conduct this survey was granted from the Ethical Review Board (ERB) of the Nepal Health Research Council (NHRC), Government of Nepal (Registration number 293/2018). Written informed consent was obtained from each participant before they enrolled in the survey. In case of minors (under 18 years old) both assent from the research participants and consent from their parents (legal guardian) was obtained, as per national ethical guidelines for health research in Nepal. We also took administrative approval from federal, provincial and local governments, as per the need. The confidentiality of all information gathered was maintained. Any waste generated during the laboratory procedures was properly disinfected

using aseptic techniques before being safely disposed of. All blood and urine samples were discarded after completing biochemical measurements.

## Results

### Characteristics of participants

Out of 6,475 participants approached for participation, 5,593 individuals participated in the study, a response rate of 86%. Just over half of the participants (53%) were female. Forty five percent (45%) of participants were aged between 15 and 29 years, with 29% aged 30 to 44 years and 26% 45to 69. Around one fifth of participants were from Lumbini Province (21%), with 19% from province 2, with the lowest proportion coming from Karnali Province (6%) and Gandaki Province (8%). Over half (57%) were from the Terai belt. Two-fifths (40%) of the participants had not completed their primary level education and approximately 46% were working as a homemaker. Just under 78% of the participants were currently married (Table 2).

### Smoking

Current smoking behavior was observed in 17% of the participants (95% CI: 15.2–19.2), with the prevalence being highest amongst men (28%; 95% CI: 24.6–31.6) and in the 45 to 69 years age group (26%; 95% CI: 22.9–28.9). The prevalence of current smoking was also higher among uneducated/less educated participants (22%; 95% CI: 18.9–24.6). There were a higher proportion of smokers found in Sudurpaschim Province (26%; 95% CI: 21.8–31.5) and Karnali Province (22%; 95% CI: 17.9–25.7) than in other provinces. Similarly, the proportion of smokers was higher in the mountain belt (27%; 95% CI: 22.4–33.0) and among the lowest quintile of affluence (poorest) (23%; 95% CI: 19.8–27.1). Conversely, the prevalence of smoking was found to be higher among employed (25%; 95% CI: 21.7–29.3) and married participants (32.3%; 95% CI: 24.8–40.9) (Table 2).

### Alcohol use

Harmful use of alcohol was observed in around 7% (95% CI: 5.5–8.4) of participants with a higher prevalence amongst males (12%; 95% CI: 10.00–15.39). Participants from the mountain belt (13%; 95% CI: 7.7–20.2) had higher prevalence compared to Terai residents (5%; 95% CI: 3.7–7.5). A higher prevalence was also observed among participants who had primary education (9%; 95% CI: 6.1–12.2) and among employed people (11%; 95% CI: 8.2–14.5) (Table 2).

### Insufficient fruit/Vegetable intake

An insufficient intake of fruits and vegetables was found among almost all participants (97%), although a slightly higher prevalence was found among those with none/less than primary education (98%; 95% CI: 95.9–99.1). Those from rural municipalities (98%; 95% CI: 97.2–99.0), the mountain belt (99%; 95% CI: 98.4–99.7) and those with the poorest economic status (99%; 95% CI: 98.8–99.8) had the highest prevalence (Table 2).

### Physical inactivity

Around 8% (95% CI: 5.7–10.1) of the participants were physically inactive with a higher prevalence among those 45 to 69 years of age (9%; 95% CI: 6.9–12.3). Participants with a primary education had a higher prevalence of physical inactivity (10%; 95% CI: 6.4–14.0) compared to participants with a secondary or higher level of education. Participants in the richest quintile (13%; 95% CI: 9.2–18.9), those who were unemployed (13%; 95% CI: 7.1–23.4) and those that

**Table 2. Prevalence of NCD risk factors among socio-demographic characteristics.**

| Characteristics of participants | Total | Current smoker | | harmful use of alcohol | | Insufficient fruit/vegetable use | | Physical inactivity) | | Overweight (%) | | Raised BP | | Raised blood sugar | | Raised cholesterol level | |
|---|---|---|---|---|---|---|---|---|---|---|---|---|---|---|---|---|---|
| | N (%) | n | % (95% CI) | n | % (95% CI) | N | % (95% CI) | n | % (95% CI) | n | % (95% CI) | n | % (95% CI) | N | % (95% CI) | n | % (95% CI) |
| **Age** | | | | | | | | | | | | | | | | | |
| 15–29 | 1466 (44.9) | 1466 | 11.7 (8.8–15.5) | 1466 | 5.3 (3.4–8.0) | 1462 | 96.4 (92.8–98.2) | 1441 | 7.8 (5.5–11.0) | 1407 | 17.2 (13.9–21.1) | 1441 | 12.9 (10.6–15.8) | 1356 | 2.48 (1.4-.5) | 1390 | 5.9 (4.3–8.0) |
| 30–44 | 2039 (28.8) | 2039 | 17.6 (14.9–20.6) | 2039 | 7.6 (5.8–9.8) | 2029 | 96.6 (94.6–97.9) | 1997 | 5.8 (3.9–8.3) | 2020 | 32.8 (29.2–36.5) | 2016 | 25.6 (22.6–28.9) | 1876 | 6.7 (4.9–9.1) | 1944 | 12 (9.8–14.6) |
| 45–69 | 2088 (26.3) | 2088 | 25.8 (22.9–28.9) | 2088 | 8.5 (6.6–10.9) | 2076 | 97.0 (94.9–98.3) | 2055 | 9.3 (6.9–12.3) | 2072 | 27.4 (23.9–31.1) | 2049 | 42.9 (39.5–46.3) | 1959 | 10.2 (8.1–12.7) | 2016 | 18.7 (16.4–21.3) |
| P-Value | | <0.001 | | 0.082 | | 0.746 | | <0.001 | | <0.001 | | <0.001 | | <0.001 | | <0.001 | |
| **Sex** | | | | | | | | | | | | | | | | | |
| Female | 3595 (53%) | 3595 | 7.5 (6.2–8.9) | 3595 | 1.75 (1.0–3.0) | 3578 | 96.3 (93.2–98.0) | 3529 | 7.3 (5.2–10.02) | 3507 | 25.1 (22.2–28.3) | 3540 | 19.7 (17.3–22.2) | 3357 | 5.3 (4.1–6.8) | 3443 | 14.0 (12.1–16.1) |
| Male | 1998 (47.0) | 1998 | 27.9 (24.6–31.6) | 1998 | 12.4 (10.0–15.4) | 1989 | 97 (94.8–98.3) | 1964 | 8.1 (5.5–11.6) | 1992 | 23.7 (20.1–27.6) | 1966 | 29.8 (26.6–33.1) | 1834 | 6.3 (4.6–8.5) | 1907 | 7.8 (6.2–9.7) |
| P-Value | | <0.001 | | <0.001 | | 0.405 | | 0.222 | | 0.454 | | <0.001 | | 0.225 | | <0.001 | |
| **Level of education*** | | | | | | | | | | | | | | | | | |
| None/less than primary | 2792 (39.7) | 2792 | 21.6 (18.9–24.6) | 2792 | 6.9 (5.2–9.3) | 2772 | 98.1 (95.9–99.1) | 2732 | 6.9 (4.9–9.5) | 2758 | 24.9 (21.9–28.3) | 2741 | 31.8 (28.7–35.1) | 2595 | 6.2 (4.8–8.1) | 2666 | 14.9 (12.8–17.3) |
| Primary | 1051 (20.1) | 1051 | 16 (11.8–21.4) | 1051 | 8.7 (6.1–12.2) | 1049 | 96.8 (93.6–98.4) | 1032 | 9.5 (6.4–14.0) | 1033 | 24.6 (20.4–29.4) | 1037 | 25.3 (21.2–29.9) | 975 | 6.5 (4.6–9.2) | 1007 | 10.42 (7.7–14.0) |
| Secondary | 1088 (24.9) | 1088 | 15.4 (12.2–19.1) | 1088 | 6.8 (4.8–9.7) | 1084 | 97.6 (95.3–98.8) | 1074 | 6.9 (4.3–11.2) | 1067 | 22.9 (18.8–27.5) | 1077 | 18.3 (14.9–22.2) | 1005 | 5.4 (3.2–8.9) | 1041 | 6.20 (4.7–8.2) |
| More than secondary | 661 (15.3) | 661 | 9.8 (6.3–14.8) | 661 | 3.78 (2.2–6.4) | 661 | 91.2 (81.7–95.9) | 654 | 8.2 (5.3–12.3) | 640 | 25.2 (19.1–32.6) | 650 | 14.7 (10.9–19.4) | 615 | 4.1 (2.3–7.2) | 635 | 10.1 (6.9–14.3) |
| P-Value | | <0.001 | | 0.079 | | <0.001 | | <0.001 | | 0.835 | | <0.001 | | 0.475 | | <0.001 | |
| **Residence** | | | | | | | | | | | | | | | | | |
| Metropolitian | 705 (8.9%) | 705 | 12.5 (7.9–19.2) | 705 | 5.3 (2.4–11.3) | 704 | 87.8 (64.9–96.5) | 699 | 6.4 (2.8–13.9) | 694 | 33.5 (26.7–41.1) | 679 | 25.2 (19.8–31.5) | 648 | 10.5 (5.3–19.6) | 668 | 9.7 (6.6–14.0) |
| Municipality | 2755 (53.8) | 2755 | 17.21 (14.9–19.8) | 2755 | 6.9 (5.3–9.1) | 2734 | 96.9 (94.1–98.4) | 2700 | 9.4 (6.6–13.1) | 2702 | 27.0 (22.9–31.6) | 2719 | 24.8 (21.9–28.0) | 2570 | 6.1 (4.4–8.5) | 2638 | 11.7 (9.8–13.9) |
| Rural municipality | 2133 (37.2) | 2133 | 18.1 (14.6–22.3) | 2133 | 6.9 (4.7–9.9) | 2129 | 98.4 (97.2–99.0) | 2094 | 5.4 (2.9–9.7) | 2103 | 18.5 (14.9–22.5) | 2108 | 23.8 (20.5–27.4) | 1973 | 4.16 (2.7–6.1) | 2044 | 10.5 (8.1–13.5) |
| P-Value | | 0.339 | | 0.809 | | <0.001 | | 0.276 | | <0.001 | | 0.855 | | 0.052 | | 0.583 | |
| **Province** | | | | | | | | | | | | | | | | | |
| Province 1 | 804 (18.3) | 804 | 10.4 (7.4–14.4) | 804 | 5.7 (3.4–9.55) | 802 | 96.4 (88.2–98.9) | 799 | 3.6 (1.6–7.9) | 790 | 25.5 (19.9–31.9) | 795 | 26.61 (21.2–32.8) | 743 | 4.40 (3.1–6.2) | 765 | 14.8 (10.8–19.8) |

(*Continued*)

**Table 2.** (Continued)

| Characteristics of participants | Total | Current smoker | | harmful use of alcohol | | Insufficient fruit/vegetable use | | Physical inactivity) | | Overweight (%) | | Raised BP | | Raised blood sugar | | Raised cholesterol level | |
|---|---|---|---|---|---|---|---|---|---|---|---|---|---|---|---|---|---|
| | N (%) | n | % (95% CI) | n | % (95% CI) | N | % (95% CI) | n | % (95% CI) | n | % (95% CI) | n | % (95% CI) | N | % (95% CI) | n | % (95% CI) |
| Province 2 | 803 (19.5) | 803 | 13.93 (10.7–17.9) | 803 | 3.72 (2.3–5.92) | 792 | 96.4 (89.5–98.8) | 796 | 8.55 (3.8–17.9) | 794 | 19.7 (14.5–26.3) | 796 | 18.7 (14.0–24.4) | 759 | 11.3 (7.4–16.9) | 770 | 11.5 (8.2–15.9) |
| Bagmati | 759 (16.2) | 759 | 18.8 (14.3–24.4) | 759 | 8.72 (5.0–14.8) | 759 | 97.2 (94.1–98.7) | 748 | 10.3 (6.8–15.3) | 755 | 42.8 (35.4–50.5) | 732 | 25.2 (20.1–31.1) | 687 | 4.1 (2.3–7.2) | 718 | 8.2 (6.2–10.7) |
| Gandaki province | 793 (8.1) | 793 | 18.9 (15.3–23.2) | 793 | 8.5 (5.2–13.6) | 791 | 98.9 (97.7–99.6) | 778 | 10.12 (4.9–19.6) | 787 | 35.4 (28.7–42.7) | 786 | 29.9 (26.6–33.5) | 757 | 3.2 (1.8–5.5) | 765 | 12.9 (9.7–17.1) |
| Lumbini Province | 797 (20.6) | 797 | 17.6 (12.5–24.1) | 797 | 7.8 (4.6–12.8) | 792 | 94.4 (82.6–98.4) | 789 | 7.3 (3.6–14.1) | 783 | 19.6 (15.8–24.2) | 780 | 28.2 (24.1–32.8) | 748 | 6.4 (3.9–10.3) | 766 | 11.6 (8.7–15.4) |
| Karnali province | 808 (5.6) | 808 | 21.6 (17.9–25.7) | 808 | 8.8 (5.7–13.3) | 806 | 96.9 (93.3–98.6) | 791 | 4.2 (1.9–8.8) | 788 | 11.4 (8.21–15.6) | 802 | 21.4 (17.2–26.3) | 763 | 0.7 (0.4–1.4) | 770 | 4.7 (3.2–6.84) |
| Sudurpaschim province | 829 (11.8) | 829 | 26.4 (21.8–31.5) | 829 | 7.0 (4.5–10.7) | 825 | 98.8 (97.7–99.4) | 792 | 9.4 (4.4–18.9) | 802 | 11.5 (8.7–15.3) | 815 | 20.9 (16.9–25.7) | 734 | 3.9 (1.5–9.7) | 796 | 9.6 (6.6–13.8) |
| P-Value | | | <0.001 | | 0.248 | | 0.504 | | 0.122 | | <0.001 | | 0.021 | | <0.001 | | 0.023 |
| **Ecological belt** | | | | | | | | | | | | | | | | | |
| Mountain | 661 (10.8) | 661 | 27.4 (22.4–33.0) | | 12.7 (7.7–20.2) | | 99.3 (98.4–99.7) | | 7.85 (5.1–11.9) | | 23.8 (17.5–31.5) | | 24.8 (18.6–32.2) | | 1.01 (0.4–2.7) | | 5.7 (3.8–8.4) |
| Hill | 2606 (31.8%) | 2606 | 16.6 (13.8–19.8) | | 7.4 (5.7–9.61) | | 97.9 (96.6–98.8) | | 6.3 (4.6–8.7) | | 31.5 (26.1–37.5) | | 27.1 (24–30.4) | | 3.0 (2.0–4.5) | | 10.4 (8.1–13.3) |
| Terai | 2326 (57.5) | 2326 | 15.5 (12.9–18.5) | | 5.3 (3.7–7.5) | | 95.4 (91.1–97.7) | | 6.7 (4.9–8.8) | | 20.6 (17.7–23.8) | | 22.9 (20.1–26.1) | | 8.2 (6.2–10.5) | | 12.5 (10.5–14.8) |
| P-Value | | | <0.001 | | <0.001 | | 0.011 | | 0.299 | | 0.001 | | 0.217 | | <0.001 | | 0.009 |
| **Wealth Quintile** | | | | | | | | | | | | | | | | | |
| Poorest | 1653 (20.0) | 1653 | 23.24 (19.78–27.11) | 1653 | 9.11 (6.59–12.46) | 1641 | 98.61 (96.98–99.37) | 1612 | 4.23 (2.55–6.95) | 1619 | 16.95 (13.67–20.83) | 1630 | 26.85 (23.27–30.75) | 1533 | 2.67 (1.61–4.41) | 1589 | 6.98 (5.30–9.15) |
| Second quintile | 1062 (20) | 1062 | 17.1 (13.9–20.9) | 1062 | 6.4 (4.7–8.7) | 1054 | 99.5 (98.8–99.8) | 1049 | 5.9 (3.8–9.3) | 1043 | 21.5 (17.9–25.5) | 1042 | 22.4 (19.0–26.3) | 998 | 4.2 (2.7–6.5) | 1020 | 10.9 (7.7–15.3) |
| Third quintile | 949 (20.1) | 949 | 15.7 (12.6–19.3) | 949 | 7.4 (4.8–11.3) | 947 | 97.9 (95.8–99.0) | 930 | 7.0 (4.1–11.7) | 928 | 22.8 (18.2–28.2) | 929 | 24.7 (20.1–29.9) | 890 | 6.5 (3.9–10.5) | 905 | 11.3 (8.5–14.8) |
| Fourth quintile | 878 (20.1) | 878 | 15.8 (12.2–20.2) | 878 | 4.7 (2.9–7.6) | 876 | 95.9 (92.4–97.8) | 868 | 7.6 (5.3–10.9) | 867 | 23.9 (19.3–29.1) | 869 | 24.5 (20.2–29.5) | 803 | 6.8 (4.2–11.1) | 833 | 12.8 (9.9–16.4) |
| Richest quintile | 1051 (19.9) | 1051 | 13.7 (10.6–17.8) | 1051 | 6.3 (4.0–9.6) | 1049 | 91.2 (83.1–95.7) | 1034 | 13.3 (9.2–18.9) | 1042 | 36.8 (30.5–43.6) | 1036 | 23.9 (19.8–28.5) | 967 | 8.7 (6.4–11.8) | 1003 | 13.5 (10.5–17.3) |
| P-Value | | | 0.002 | | 0.177 | | <0.001 | | 0.001 | | <0.001 | | <0.001 | | 0.008 | | 0.032 |
| **Occupation***  | | | | | | | | | | | | | | | | | |
| Employed | 1707 (32.9) | 1707 | 25.3 (21.7–29.3) | 1707 | 10.9 (8.2–14.5) | 1700 | 96.5 (93.8–98.0) | 1685 | 8.7 (6.0–12.6) | 1689 | 27.5 (23.6–31.9) | 1687 | 31.6 (28.1–35.3) | 1566 | 6.5 (5.1–8.3) | 1625 | 11.2 (8.5–14.4) |

(*Continued*)

**Table 2.** (Continued)

| Characteristics of participants | Total | Current smoker | | harmful use of alcohol | | Insufficient fruit/vegetable use | | Physical inactivity) | | Overweight (%) | | Raised BP | | Raised blood sugar | | Raised cholesterol level | |
|---|---|---|---|---|---|---|---|---|---|---|---|---|---|---|---|---|---|
| | N (%) | n | % (95% CI) | n | % (95% CI) | N | % (95% CI) | n | % (95% CI) | n | % (95% CI) | n | % (95% CI) | N | % (95% CI) | n | % (95% CI) |
| Student | 402 (14.3) | 402 | 3.6 (1.7–7.3) | 402 | 1.66 (0.6–4.5) | 400 | 95.1 (88.2–98.1) | 396 | 6.3 (3.6–10.8) | 393 | 12.3 (7.6–19.3) | 393 | 6.6 (3.8–11.2) | 374 | 1.7 (.73–4.1) | 386 | 3.8 (2.1–6.9) |
| Homemaker | 3142 (45.5) | 3142 | 15.2 (12.9–17.7) | 3142 | 5.3 (3.69–7.45) | 3131 | 97.4 (95.7–98.4) | 3080 | 6.4 (4.6–8.8) | 3076 | 25.8 (22.6–29.3) | 3090 | 24.9 (21.9–28.0) | 2927 | 6.3 (4.5–8.8) | 3009 | 13.3 (11.5–15.4) |
| Unemployed | 273 (6.1) | 273 | 20.9 (12.5–32.8) | 273 | 8.4 (4.6–15.0) | 267 | 95.4 (89.7–98.0) | 263 | 13.3 (7.1–23.4) | 272 | 24.0 (14.2–37.6) | 269 | 23.4 (17.3–30.9) | 256 | 5.4 (2.49–1.4) | 261 | 10.6 (6.41–17.1) |
| Others | 63 (0.9) | 63 | 11.4 (5.1–23.5) | 63 | 2.3 (0.80–6.5) | 63 | 98.7 (92.6–99.8) | 63 | 12.22 (4.7–28.2) | 63 | 27.7 (14.3–46.7) | 61 | 34.3 (20.7–51.1) | 62 | 16.9 (8.0–32.1) | 63 | 16.3 (7.5–31.9) |
| P-Value | | | <0.001 | | <0.001 | | 0.048 | | <0.001 | | 0.002 | | <0.001 | | 0.002 | | <0.001 |
| **Marital status*** | | | | | | | | | | | | | | | | | |
| Unmarried | 538 (19.5) | 538 | 10.0 (6.6–15.0) | 538 | 4.0 (1.8–8.7) | 534 | 96.5 (92.4–98.4) | 531 | 8.7 (5.4–13.5) | 531 | 13.79 (9.5–19.6) | 530 | 12.7 (9.0–17.7) | 496 | 1.71 (0.7–4.1) | 509 | 4.43 (2.6–7.4) |
| Currently married | 4752 (77.8) | 4752 | 18.4 (16.4–20.5) | 4752 | 7.4 (6.1–9.1) | 4735 | 96.6 (94.4–97.9) | 4666 | 7.2 (5.3–9.6) | 4668 | 27.4 (24.5–30.5) | 4675 | 26.9 (24.7–29.2) | 4412 | 6.7 (5.3–8.5) | 4552 | 12.3 (10.6–14.2) |
| Separated/ Divorced/ Widowed | 302 (2.7) | 302 | 32.3 (24.8–40.9) | 302 | 8.20 (3.7–17.1) | 297 | 98.0 (93.7–99.4) | 295 | 13.2 (7.9–21.3) | 299 | 15.7 (10.7–22.3) | 300 | 40.6 (33.2–48.5) | 282 | 6.6 (3.8–11.5) | 288 | 22.9 (17.0–30.1) |
| P-Value | | | <0.001 | | 0.155 | | 0.793 | | <0.001 | | <0.001 | | <0.001 | | <0.001 | | <0.001 |
| **Total** | | **5593** | **17.1 (15.2–19.2)** | **5593** | **6.8 (5.5–8.4)** | **5567** | **96.6 (94.3–98.0)** | **5493** | **7.7 (5.7–10.1)** | **5499** | **24.4 (21.7–27.4)** | **5506** | **24.5 (22.4–26.7)** | **5191** | **5.8 (4.5–7.3)** | **5350** | **11.1 (9.7–12.7)** |

* 1 case from education, 6 cases from occupation and 1 case from marital status was excluded.

were married (13%; 95% CI: 7.9–21.3) had a higher proportion of physical inactivity as compared to their counterparts (Table 2).

## Overweight

The prevalence of overweight was 24% (95% CI: 21.7–27.4) across all participants, a higher prevalence was found among participants in the 30 to 44 years age group (33%; 95% I: 29.2–36.5). Metropolitan city residents (34%; 95% CI: 26.7–41.1) and hill residents (32%; 95% CI: 26.1–37.5) had a higher prevalence of overweight compared to residents in rural municipality (18%; 95% CI: 14.9–22.5) and Terai belt (21%; 95% CI: 17.7–23.8]. The highest prevalence of overweight was found in Bagmati Province (43%; 95% CI: 35.4–50.5) followed by Gandaki Province (35%; 95% CI: 28.7–42.7). Similarly, the prevalence was highest amongst those in the richest quintile (37%; 95% CI: 30.5–43.6) and currently married participants (27%; 95% CI: 24.5, 30.5).

## Raised BP

Around 24% (95% CI: 22.4–26.7) of the participants had raised BP with a higher prevalence among men (30%; 95% CI: 26.6–33.1) and participants in the 45 to 69 years age group (43%; 95% CI: 39.5–46.3). Raised BP was most prevalent in Gandaki (30%; 95% CI: 26.6, 33.5) and

Lumbini Provinces (28%; 95% CI: 24.1–32.8) as compared to other provinces. Similarly, a higher prevalence was observed among participants having none/less than primary level of education (32%; 95% CI: 28.7–35.1) and those that were married (41%; 95% CI: 33.2–48.5).

## Raised blood sugar

The prevalence of raised blood sugar was 5.8% for the total sample (95% CI: 4.5–7.3). Around 10% of participants aged 4 to 69 years (10.2%; 95% CI: 8.1–12.7) and 9% of participants in the richest quintile (95% CI: 6.4–11.8) had raised blood sugar. The highest regional prevalence was observed among participants of province 2 (11%; 95% CI: 7.4–16.9) with the lowest in Karnali Province (1%; 95% CI: 0.36–1.4). Likewise, metropolitan (10%; 95% CI: 5.3–19.6) and Terai residents (8%; 95% CI: 6.2–10.7) had higher prevalence of raised blood sugar compared to rural municipality (4%; 95% CI: 2.7–6.1) and hilly areas (3%; 95% CI: 2.0–4.5) (Table 2).

## Raised cholesterol level

Raised cholesterol level was found among 11% of the participants (95% CI: 9.7–12.7), with this highest among participants 45 to 69 years of age (19%; 95% CI: 16.4–21.3) and among females (14%; 95% CI: 12.11–16.17). Compared to other levels of education, participants with 'none/less than primary education' (15%; 95% CI: 12.7–17.3) had the highest prevalence of raised total cholesterol. Whilst a higher prevalence was found in Province 1 (15%; 95% CI: (10.81–19.84), Terai residents (12%; 95% CI: 10.5, 14.8), richest quintile (14%; 95% CI: 10.5–17.3) and married participants (23%; 95% CI: 17.0–30.1) compared to their counterparts (Table 2).

Prevalence ratios demonstrated a significantly higher prevalence of smoking among males (APR: 4.49, 95% CI: 3.70–5.46) compared to females, once adjusting for other covariates. Smoking was significantly lower among participants having more than a secondary level education (APR: 0.56, 95% CI: 0.39–0.81) compared to participants with no, or less than primary level of education. Similarly, a lower prevalence was found within Province 1 (APR: 0.42, 95% CI: 0.29–0.60) residents and participants of hilly region (APR: 0.69, 95% CI: 0.54–0.90) as compared to reference categories of Sudurpaschim Province and those from the mountain region, respectively (Table 3).

Likewise, alcohol intake was significantly higher among men (APR: 9.09, 95% CI: 5.38–15.35) and lower among participants having more than a secondary level of education (APR: 0.5, 95% CI: 0.28–0.9) and those residing in the Terai region (APR: 0.38, 95% CI: 0.20–0.70) (Table 3).

Insufficient intake of fruits and vegetables was significantly less prevalent among participants having more than a secondary level education (APR: 0.94, 95% CI: 0.88–1.00) and among participants of Karnali Province (APR: 0.97, 95% CI: 0.94–1.00) than Sudurpaschim residents. A higher prevalence was observed among participants in the second poorest quintile (APR: 1.02, 95% CI: 1.0–1.03) compared to those in the poorest quintile. Similarly, low physical activity was significantly lower among participants of Province 1 (APR: 0.3, 95% CI: 0.10–0.83) and higher among richest participants (APR: 2.74, 95% CI: 1.42–5.27) (Table 3).

Being overweight was significantly higher among participants aged 30 to 44 years (APR: 1.46, 95% CI: 1.18–1.80) compared to those aged 15 to 29 years. A higher prevalence was observed among participants of Bagmati Province (APR: 2.5, 95% CI: 1.82–3.49) and Gandaki Province (APR: 2.36, 95% CI: 1.71–3.26) and lower among Terai residents (APR: 0.70, 95% CI: 0.53–0.94) than participants of mountain area. Similarly, raised BP (APR: 2.52, 95% CI: 2.06–3.09) and raised blood sugar (APR: 3.9, 95% CI: 2.05–7.36) was significantly higher among participants within the 45 to 49 years age group. A higher prevalence of raised BP was found amongst males (APR: 1.5, 95% CI: 1.27–1.77) and participants of Gandaki Province (APR: 1.3,

**Table 3. Adjusted prevalence ratio of sociodemographic characteristics with NCD risk factors.**

| | Smoking (APR with 95% CI) | Harmful use of alcohol (APR with 95% CI) | Insufficient fruit/ vegetable intake (APR with 95% CI) | Physical inactivity (APR with 95% CI) | Overweight (APR with 95% CI) | Raised BP (APR with 95% CI) | Raised Sugar (APR with 95% CI) | Raised blood cholesterol (APR with 95% CI) |
|---|---|---|---|---|---|---|---|---|
| **Age** | | | | | | | | |
| 15–29 | Ref | Ref | Ref | Ref | Ref | Ref | Ref | Ref |
| 30–44 | 1.1 (0.83–1.44) | 0.94 (0.61–1.46) | 0.99 (0.97–1.02) | 0.72 (0.46–1.12) | 1.46 (1.18–1.80)*** | 1.61 (1.30–1.98)*** | 2.33 (1.30 -.19)** | 1.69 (1.15–2.48)** |
| 45–69 | 1.39 (1.03–1.87)* | 0.94 (0.59–1.50) | 0.99 (0.96–1.01) | 1.14 (0.76–1.69) | 1.26 (1.04–1.53)* | 2.52 (2.06 -.09)*** | 3.88 (2.05 -.36)*** | 2.58 (1.75 -.78)*** |
| **Sex** | | | | | | | | |
| Female | Ref | Ref | Ref | Ref | Ref | Ref | Ref | Ref |
| Male | 4.49 (3.70–46)*** | 9.09 (5.38 -.35)*** | 0.98 (0.97–1.00) | 0.91 (0.60–1.39) | 0.95 (0.81–1.12) | 1.5 (1.27 -.77)*** | 1.04 (0.67–1.61) | 0.51 (0.38 -.67)*** |
| **Education level** | | | | | | | | |
| None/less than primary | Ref | Ref | Ref | Ref | Ref | Ref | Ref | Ref |
| Primary | 0.8 (0.63–1.02) | 1.15 (0.81–1.64) | 0.98 (0.96–1.00) | 1.39 (0.90–2.14) | 0.99 (0.82–1.19) | 0.95 (0.82–1.10) | 1.68 (1.16–2.43)** | 0.89 (0.65–1.20) |
| Secondary | 0.8 (0.63–1.02) | 0.85 (0.54–1.34) | 1 (0.97–1.02) | 0.91 (0.57–1.46) | 0.93 (0.75–1.15) | 0.79 (0.63–1.00)* | 1.65 (1.03–2.65)* | 0.66 (0.47–0.91)* |
| more than secondary | 0.56 (0.39–0.81)** | 0.5 (0.28–0.90)* | 0.94 (0.88–1.00)* | 0.83 (0.45–1.54) | 0.91 (0.71–1.19) | 0.66 (0.48–0.92)* | 1.29 (0.64–2.62) | 1.06 (0.67–1.69) |
| **Residence** | | | | | | | | |
| Rural municipality | Ref | Ref | Ref | Ref | Ref | Ref | Ref | Ref |
| (Sub) Metropolitan | 0.79 (0.53–1.17) | 0.85 (0.40–1.84) | 0.93 (0.81–1.07) | 0.54 (0.21–1.37) | 1.22 (0.93–1.59) | 1.08 (0.85–1.36) | 1.46 (0.72–2.97) | 0.73 (0.50–1.07) |
| Municipality | 1.02 (0.82–1.26) | 0.88 (0.56–1.39) | 1.01 (0.99–1.03) | 0.78 (0.40–1.52) | 0.78 (0.63–0.96)* | 0.91 (0.77–1.09) | 0.79 (0.48–1.31) | 1 (0.75–1.32) |
| **Province** | | | | | | | | |
| Sudurpaschim | Ref | Ref | Ref | Ref | Ref | Ref | Ref | Ref |
| Province 1 | 0.42 (0.29–60)*** | 1.01 (0.55–1.87) | 0.98 (0.94–1.03) | 0.3 (0.10–0.83)* | 1.98 (1.38–2.84)*** | 1.23 (0.92–1.63) | 0.68 (0.25–1.84) | 1.16 (0.74–1.82) |
| Province 2 | 0.53 (0.38 -.75)*** | 0.75 (0.39–1.44) | 1 (0.96–1.04) | 0.6 (0.21–1.76) | 1.51 (1.03–2.21)* | 0.82 (0.59–1.15) | 1.45 (0.50–4.25) | 0.76 (0.47–1.21) |
| Bagmati Province | 0.67 (0.51–0.89)** | 1.1 (0.56–2.16) | 1 (0.97–1.03) | 0.84 (0.38–1.84) | 2.52 (1.82–3.49)*** | 0.95 (0.69–1.31) | 0.79 (0.29–2.14) | 0.63 (0.39–1.02) |
| Gandaki Province | 0.79 (0.60–1.04) | 1.37 (0.70–2.68) | 1 (0.98–1.03) | 0.89 (0.33–2.41) | 2.36 (1.71–3.26)*** | 1.3 (1.00–1.68)* | 0.67 (0.22–2.01) | 1.02 (0.66–1.60) |
| Lumbini Province | 0.72 (0.52–0.98)* | 1.57 (0.87–2.86) | 0.97 (0.92–1.03) | 0.56 (0.22–1.42) | 1.59 (1.13–2.23)** | 1.34 (1.03–1.74)* | 1 (0.34–2.94) | 0.9 (0.58–1.40) |
| Karnali Province | 0.9 (0.67–1.21) | 1.31 (0.74–2.32) | 0.97 (0.94–1.00*) | 0.51 (0.17–1.49) | 0.94 (0.61–1.43) | 1.02 (0.76–1.37) | 0.26 (0.08 -.81)* | 0.53 (0.31 -.94)* |
| **Ecological** | | | | | | | | |
| Mountain | Ref | Ref | Ref | Ref | Ref | Ref | Ref | Ref |
| Hill | 0.69 (0.54 -.90)** | 0.53 (0.30–0.95)* | 1.01 (0.98–1.03) | 2.32 (0.88–6.09) | 1.03 (0.80–1.32) | 0.94 (0.69–1.27) | 2.46 (0.81–7.49) | 1.53 (0.94–2.47) |
| Terai | 0.7 (0.53–0.92)** | 0.38 (0.20–0.70)** | 0.98 (0.96–1.01) | 2.76 (0.95–8.05) | 0.7 (0.53–0.94)* | 0.77 (0.57–1.04) | 4.25 (1.35–13.36)* | 1.61 (0.94–2.77) |
| **Wealth Quintile** | | | | | | | | |
| Poorest quintile | Ref | Ref | Ref | Ref | Ref | Ref | Ref | Ref |
| Second quintile | 0.88 (0.71–1.08) | 0.84 (0.53–1.34) | 1.02 (1.00–1.03)* | 1.27 (0.72–2.27) | 1.26 (1.02–1.56)* | 0.88 (0.74–1.06) | 1.2 (0.65–2.19) | 1.49 (1.08–2.07)* |

*(Continued)*

**Table 3.** (Continued)

| | Smoking (APR with 95% CI) | Harmful use of alcohol (APR with 95% CI) | Insufficient fruit/ vegetable intake (APR with 95% CI) | Physical inactivity (APR with 95% CI) | Overweight (APR with 95% CI) | Raised BP (APR with 95% CI) | Raised Sugar (APR with 95% CI) | Raised blood cholesterol (APR with 95% CI) |
|---|---|---|---|---|---|---|---|---|
| Third quintile | 0.84 (0.66–1.08) | 1.16 (0.71–1.90) | 1.01 (0.99–1.03) | 1.31 (0.65–2.66) | 1.42 (1.07–1.88)* | 1.06 (0.83–1.37) | 1.4 (0.70–2.80) | 1.54 (1.01–2.37)* |
| Fourth quintile | 0.88 (0.66–1.17) | 0.7 (0.37–1.30) | 1 (0.97–1.02) | 1.5 (0.82–2.74) | 1.51 (1.18–1.94)** | 1.06 (0.86–1.30) | 1.2 (0.58–2.47) | 1.86 (1.25–2.75)** |
| Richest quintile | 0.87 (0.65–1.17) | 1.08 (0.59–1.97) | 0.96 (0.92–1.00) | 2.74 (1.42–5.27)** | 1.94 (1.52–2.50)*** | 1.07 (0.85–1.35) | 1.5 (0.72–3.16) | 2.09 (1.38–3.19)*** |
| **Occupation** | | | | | | | | |
| Employed | Ref | Ref | Ref | Ref | Ref | Ref | Ref | Ref |
| Student | 0.2 (0.09 -.45)*** | 0.24 (0.08–0.68)** | 0.97 (0.92–1.02) | 0.59 (0.26–1.33) | 0.74 (0.43–1.25) | 0.3 (0.17 -.56)*** | 1.06 (0.25–4.55) | 0.57 (0.28–1.18) |
| Homemaker | 1.08 (0.88–1.32) | 1.17 (0.78–1.75) | 1 (0.99–1.02) | 0.8 (0.59–1.10) | 1.03 (0.87–1.21) | 0.87 (0.72–1.06) | 1.09 (0.66–1.79) | 0.84 (0.58–1.20) |
| Unemployed | 0.9 (0.57–1.42) | 0.91 (0.49–1.70) | 0.99 (0.94–1.03) | 1.65 (0.88–3.08) | 0.83 (0.59–1.16) | 0.71 (0.51–0.98)* | 1.19 (0.55–2.56) | 0.85 (0.54–1.34) |
| Others | 0.39 (0.19–0.81)* | 0.25 (0.09–0.70)** | 1.03 (0.99–1.07) | 1.14 (0.44–2.94) | 0.85 (0.51–1.43) | 0.86 (0.58–1.28) | 2.25 (1.16 -.37)* | 1.2 (0.60–2.41) |
| **Marital status** | | | | | | | | |
| Unmarried | Ref | Ref | Ref | Ref | Ref | Ref | Ref | Ref |
| Currently married | 0.85 (0.58–1.25) | 1.13 (0.53–2.40) | 0.98 (0.94–1.02) | 0.72 (0.41–1.25) | 1.28 (0.88–1.85) | 0.73 (0.49–1.09) | 1.92 (0.54–6.78) | 1.13 (0.59–2.15) |
| Separated/ Divorced/ Widowed | 1.68 (0.98–2.90) | 1.77 (0.66–4.70) | 0.99 (0.94–1.03) | 1.26 (0.56–2.85) | 0.75 (0.45–1.24) | 0.89 (0.57–1.40) | 1.55 (0.37–6.41) | 1.4 (0.68–2.91) |

* $p < 0.05$;

** $p < 0.01$;

*** $p < 0.001$.

95% CI: 1.0–1.7) and Lumbini Province (APR: 1.3, 95% CI: 1.03–1.74). Whilst raised blood sugar was significantly higher among Terai residents (APR: 4.3, 95% CI: 1.35–13.36) (Table 3).

A significantly higher prevalence of raised blood cholesterol was observed among participants within the 45 to 69 years age group (APR: 2.58, 95% CI: 1.75–3.78). This was lower among men (APR: 0.51, 95% CI: 0.38–0.67) and participants from the Karnali Province (APR: 0.53, 95% CI: 0.31–0.94) compared to females and Sudurpaschim residents, respectively. A higher prevalence was found among those in the most affluent quintile (APR: 2.09, 95% CI: 1.38–3.19) (Table 3).

Age, sex, education, residence, province and wealth were significantly associated with clustering of risk factors. Males (ARR: 1.2, 95% CI: 1.1–1.3) and those in the fourth wealth quintile (ARR: 1.17, 95% CI: 1.07–1.28) had a significantly higher number of risk factors compared to females and the poorest participants. Similarly, participants who had more than a secondary level education (ARR: 0.86, 95% CI: 0.78–0.95) and those who resided in Karnali Province (ARR: 0.9, 95% CI: 0.8–0.9) had fewer risk factors (Table 4).

## Discussion

### Smoking

The prevalence of current smoking (17.1%) is relatively stable from the previous round of the STEPS survey (19%) and this findings is similar to that of Bangladesh's GATS 2017 survey

**Table 4. Clustering of NCD risk factors and its multivariable analysis.**

| Age | Mean number of existing risk factors (95% CI) | Adjusted relative risk ARR (95% CI) |
|---|---|---|
| 15–29 years | 1.81 (1.75–1.86) | Ref |
| 30–44 years | 2.00 (1.96–2.05) | 1.14 (1.06–1.22)*** |
| 45–69 years | 1.95 (1.91–1.98) | 1.31 (1.23–1.39)*** |
| **Sex** | | |
| Female | 1.95 (1.91–1.98) | Ref |
| Male | 2.41 (2.36–2.46) | 1.21 (1.14–1.29)*** |
| **Education level** | | |
| None/less than primary | 2.15 (2.11–2.20) | Ref |
| Primary | 2.11 (2.04–2.18) | 0.99 (0.94–1.04) |
| Secondary | 2.09 (2.02–2.16) | 0.94 (0.92–1.01) |
| more than secondary level | 1.95 (1.86–2.04) | 0.86 (0.78–0.95)** |
| **Residence** | | |
| Rural municipality | 1.99 (1.94–2.03) | Ref |
| Sub/metropolitan | 2.30 (2.21–2.39) | 0.94 (0.81–1.09) |
| Municipality | 2.16 (2.11–2.21) | 0.95 (0.89–1.01) |
| **Province** | | |
| Sudurpaschim | 2.09 (2.01–2.8) | Ref |
| Province 1 | 2.15 (1.91–2.07) | 0.91 (0.82–1.02) |
| Province 2 | 1.99 (1.91–2.08) | 0.89 (0.78–1.01) |
| Bagmati Province | 2.29 (2.20–2.37) | 0.98 (0.90–1.07) |
| Gandaki Province | 2.29 (2.21–2.38) | 1.07 (0.97–1.18) |
| Lumbini Province | 2.07 (1.99–2.15) | 0.95 (0.86–1.04) |
| Karnali Province | 1.90 (1.83–1.97) | 0.88 (0.80–0.96)** |
| **Ecological** | | |
| Mountain | 2.08 (1.99–2.15) | Ref |
| Hill | 2.13 (2.09–2.17) | 0.99 (0.92–1.06) |
| Terai | 2.10 (2.05–2.15) | 0.93 (0.85–1.01) |
| **Wealth Quintile** | | |
| Poorest | 1.95 (1.91–2.00) | Ref |
| Second quintile | 2.05 (1.99–2.12) | 1.02 (0.96–1.09) |
| Third quintile | 2.11 (2.04–2.18) | 1.06 (0.98–1.14) |
| Fourth quintile | 2.20 (2.12–2.29) | 1.1 (1.02–1.18)* |
| Richest | 2.33 (2.25–2.41) | 1.17 (1.07–1.28***) |
| **Occupation** | | |
| Employed | 2.28 (2.21–2.33) | Ref |
| Student | 2.22 (2.10–2.35) | 0.75 (0.68–0.84)*** |
| Homemaker | 1.99 (1.96–2.04) | 0.98 (0.91–1.04) |
| Unemployed | 2.13 (1.99–2.27) | 0.95 (0.86–1.05) |
| Others | 2.5 (2.21–2.79) | 0.92 (0.77–1.09) |
| **Marital status** | | |
| Unmarried | 2.13 (2.03–2.23) | Ref |
| Currently married | 2.09 (2.07–2.13) | 0.93 (0.84–1.03) |
| Separated/Divorced/ Widowed | 2.29 (2.17–2.42) | 0.99 (0.88–1.12) |
| **Total** | **2.04 (2.02–2.08)** | - |

* $p < 0.05$;

** $p < 0.01$;

*** $p < 0.001$.

[10,18,19]. However, compared to India's smoking prevalence (10.7%), the prevalence in Nepal is higher [20]. This relatively stable smoking rate from 2013 onward, could be the result of implementation and monitoring of the comprehensive tobacco control law that was introduced in 2011 [21]. Furthermore, an increase in literacy rate of the population, increased awareness about the health consequences of smoking, effective implementation and monitoring of tobacco control law provisions such as pictorial health warning, tobacco industry litigation, may have played a crucial role in keeping the smoking prevalence stable, or to curb the increasing trend.

Within our study we found a significantly higher smoking prevalence amongst males (25%) than females (7%), which aligns with the patterns of smoking observed in WHO SEARO member countries [22]. Studies have indicated that this could due to a range of factors including tobacco industry market strategies that portray smoking as more masculine and community tolerance of male smoking over female smoking [23,24]. Our findings also found an increasing prevalence in smoking with increasing age, a similar finding to that of the previous 2013 STEPS survey and other global data [10,22]. A possible explanation for increasing smoking prevalence with age may be due to increased levels of dependence with age, or lack of effective cessation programs which may lead to the accumulation of smokers with increasing age [25].

We found that participants residing in Province 1, Province 2, Bagmati Province, and Lumbini Province were less likely to smoke than those in Sudurpaschim Province, with this finding aligning with another national level survey [26]. This may be due to the comparatively high levels of literacy in those provinces compared to Karnali, Gandaki and Sudurpaschim. The role of education in smoking practices is further elucidated through the relationship of educational level and prevalence of smoking found within the current study. Participants with none/less than primary of education were more likely to smoke as compared to those with a higher education level (primary, secondary or higher secondary above) in our study. This finding was consistent with previous data from demographic and health surveys of nine countries, including Nepal [27]. Similarly, people residing in mountainous region were more likely to smoke than in any other regions of country and these findings, which aligns with previous survey findings [10]. The differences in prevalence of smoking based on province and ecological belt could indicate the need of contextualized targeted interventions for smoking control in Nepal. The current federal structure of the country, where planning process is devolved to provincial and local government to a large extent, could be an opportunity for implementation of locally contextualized interventions for control of smoking.

## Alcohol intake

Prevalence of harmful alcohol intake has increased to 6.7% in the current study from the 2.2% reported in 2013 STEPS survey [10]. Alcohol intake and harmful alcohol intake was higher among males than females, which was also noted in the previous round of STEPS survey [10]. Regarding types of alcohol used, a significant proportion of females consumed home-brewed alcohol whilst males consumed alcohol from other sources i.e. industrially produced alcohol [28]. This difference in consumption of alcohol based on the sex of participants could be linked to social and cultural norms which define drinking alcohol by males as normal behavior, while in females, drinking alcohol is still considered as an anti-social act [24]. However, compared to previous rounds of the STEPS survey, a higher proportion of females are consuming alcohol, which could be a result of changing lifestyles and societal perceptions in alcohol consumption among females. In addition, findings revealed that there is a higher prevalence of the harmful use of alcohol among employed participants (10.96%) compared to other groups.

Our study revealed participants with higher education level (secondary, more than secondary) and Terai residents were less likely to consume harmful levels of alcohol. This finding is in line with a previous study conducted among 9,000 females, in which those of the mountain region and those having no education/formal education were more likely to drink alcohol [29]. This may be due to socio-cultural differences among different ecological belts of Nepal. In the majority of ethnic groups in the mountainous region there is a cultural acceptance of drinking alcohol, whereas in the Terai region drinking alcohol is considered an unreligious act [29].

## Insufficient fruit and vegetable intake

The results of fruit and vegetables intake suggests that there is marginal improvement intake in comparison to previous round of STEPS survey [10]. Multivariable analysis found no significant association with achieving recommended levels of fruit and vegetable intake. However, this study has found participants with higher education level (more than secondary level of education) participants were more likely to consume adequate levels of fruits and vegetables when compared with less educated groups, similar to findings of a previous study [30]. In the context of Nepal, factors such as limited accessibility, availability and affordability of fruits and vegetables and social perceptions on the use of fruits and vegetables could have a role in the high prevalence of an insufficient intake of fruit and vegetables in the population. Individuals may also lack adequate information on the need to consume sufficient fruit and vegetables and the health consequences of insufficient intake. This issue could be further explored through qualitative research, which could provide more in-depth insights into the insufficient fruit and vegetable intake among the Nepalese population. Findings from such studies could also be useful in designing contextualized interventions intended to promote adequate intake of fruits and vegetables.

## Physical inactivity

The current study reports a low prevalence of physical inactivity (7.4%) a finding that is in line with those of previous national and international surveys [31,32]. However, in comparison with the 2013 STEPS survey, physical inactivity has doubled [10]. Those in the richest quintile were found to have the highest prevalence of physical inactivity. This may be due to the adoption of a sedentary lifestyle associated with occupations among this group of people [33] along with better access to means of transportation, thereby reducing walking hours in a day.

## Overweight

Almost of one quarter (24%) of people were overweight, a figure slightly higher than that reported in STEPS survey in 2013 (21%) [10]. This increment may be understood in relation to changes in physical inactivity level, which was about 3% in 2013 and has increased to 7.4%. Apart from sedentary lifestyle, urbanization accompanied with increased consumption of processed/junk foods may be a factor in the increased prevalence of overweight among Nepalese adults.

There is increasing prevalence of overweight with increasing age group, a finding in line with previous publications from the 2013 STEPS survey. Ageing may also be associated with limited mobility and limited engagement in labor intensive works which could result in overweight among participants of relatively higher age group. Those in the richest quintile have higher a prevalence of overweight compared to those in the poorest quintile which is similar to findings from a systematic review on the South Asian context [34]. A higher prevalence of overweight among females could be attributed to social and cultural factors which influence

both dietary intake and physical activity [35]. However, this finding was not supported by multivariable analysis in the present study.

## Raised BP

Within the present study one quarter (24.44%) of Nepalese had raised BP, which is consistent with previous round of STEPS survey, but slightly higher than reported in the 2016 NDHS survey (19.9%) [36]. This difference may be due to methodological variation i.e. differences in sampling design. Findings from both surveys indicate an increasing burden of raised BP in Nepal and demand sufficient efforts for prevention and control of this problem.

We found an increasing prevalence of raised BP with increasing age, which is similar to the previous STEPS survey and the Nepal Demographic and Health Survey (NDHS) 2016. We also found a sex difference in the prevalence of raised BP, which is consistent with other surveys' findings [10,36]. Sex differences in raised BP may be due to both biological and behavioral factors [37]. Such as sex hormones, genetic makeup, and other biological sex features that are assumed to have a protective effect against raised BP in females [37,38]. An association was also found between education level raised BP, with a lower prevalence found amongst more educated participants. This result is consistent with the findings of previous STEPS and NDHS. Educated people are likely to have access to information about the raised BP and its consequences, which might ultimately help them to adopt preventative measures [39].

## Raised blood sugar

WHO global estimates has shown that 8% of South Asian people have increased level of blood sugar level which is close to the estimates in this study (6%) [40]. The prevalence of raised blood sugar has doubled from 3% to 6% [10] which should also be interpreted considering the difference in techniques to measure blood sugar level. In previous rounds of the survey the wet method was adopted to measure blood sugar level, however, for this round of survey the dry method was used. The prevalence of raised blood sugar level increased with increasing age group, which is comparable to other national surveys. Increasing age is associated with combined effect of increasing adiposity, decreasing physical activity, medications, coexisting illness, and insulin secretary defects that effect blood sugar level [41]. Similarly, this study has reported differences in prevalence of raised blood sugar between provinces, place of residence (sub/metropolitan city, municipality or rural municipality), and ecological region (mountain, hill, Terai). Furthermore, blood sugar difference among ecological region is further validated by multivariable analysis, that has shown that residents from Terai are more likely to have raised blood sugar compared to those from the mountainous region. These differences may be attributed to variations in physical activity levels, dietary habits and urbanization level, with this findings similar to that from previous studies [42,43]. Some of the studies have put forward a biological explanation that increased content of the glucose transporter GLUT4 in the plasma membrane of skeletal muscle cells incubated under anoxia conditions (35,38), and in skeletal muscle cells exposed to prolonged hypoxia leads to the better glucose tolerance [44,45].

## Raised cholesterol level

The prevalence of raised total cholesterol as found in the current study is quite low (11%) compared with previous rounds of the survey i.e 22.7%; this may be due to differences in measurement techniques. As with raised BP, we found an increase in prevalence of raised cholesterol level with increasing age group, with this finding similar to that of the 2013 survey. Reduction in the production of growth hormone with increasing age may be a causal factor contributing

to the age-dependent rise in blood cholesterol [46]. Similarly, our finding that females have a higher prevalence of raised blood cholesterol level may be linked to increasing age and fluctuations in female sex hormone i.e. estrogen. Various studies have shown that estrogen helps to maintain levels of high-density lipoproteins (HDL) in adult females. However, at menopause many females experience a change in their cholesterol levels, with total cholesterol and low-density lipoproteins (LDL) levels rising and HDL falling [47]. In addition, a greater prevalence of raised total cholesterol in participants in the richest quintile, as found in the present study, may be due to the adoption of a more sedentary lifestyle, a lack of physical activity and stress related factors.

## Clustering of risk factors

The current study reveals that Nepalese adults on average have the presence of two risk NCD risk factors. With the average number of NCD risk factors greater in males, the richest wealth quintile, and amongst older participants. Suggesting that increasing age is associated with increasing clustering of risk factors, a finding supported by research from other countries [48,49]. As Nepal has been experiencing a rapid increase in life expectancy and median age of population, it is likely that such problems could escalate in the coming years [1]. Greater clustering of risk factors in males compared to females may be due to risk-oriented behavior and sedentary lifestyle in male such as tobacco smoking, alcohol and physical inactivity.

Our finding that the prevalence of clustering of NCDs risk factors is higher among the richest, was also found in a previous study in Bhutan [50]. Similar to individual risk factors such as overweight/obesity and hypertension, the clustering of NCDs risk factors in the richest group can be linked with the adoption of a sedentary lifestyle.

## Policy implications and way forward

The policies and programs targeted to reduce NCD risk factors within the Nepal population should be designed as per the socio-demographic gradient of the country. Finally, the new multi-sectoral action plan for prevention and control of NCDs in Nepal should consider the federal context and trends of risk factors for effective prevention and control in Nepal.

## Conclusion

The findings for this survey demonstrate that a large proportion of the Nepalese population is living with two or more NCD risk factors. In comparison to the 2013 STEPS survey, prevalence of most of the risk factors has increased, indicating a need for effective programs to counter this. One of the primary strategies to reduce the burden of NCD risk factors would be to prevent, or reduce, the burden of modifiable risk factors, which could also prove more cost effective than providing curative services to people with NCDs. However, interventions on modifiable risk factors demand collaborative efforts from multiple sectors so as to create an enabling environment for behavior change. The current federal structure in Nepal, in which the municipality takes responsibility for different sectors like education, infrastructure development, environment etc. together with health, can provide an opportunity for integrated interventions from different sectors, which could prove effective in reducing the burden of NCD risk factors in the country.

## Supporting information

**S1 File.**
(PDF)

**S1 Dataset.**
(XLSX)

## Acknowledgments

We would like to acknowledge the effort of all the individuals involved in this survey, express our deep sense of appreciation to the steering committee and technical working group (TWG) members. We are grateful to World Health Organization (WHO) for technical support to conduct this survey. In particular, we would like to express my sincere thanks to Dr. Manju Rani and Naveen Agarwal (HO/SEARO); Dr. Patricia Rarau and Dr. Stefan Savin (WHO HQ); Dr. Md. Khurshid Alam Hyder and Dr. Lonim Prasai Dixit (WHO Nepal); Ms. Yvonne Y. Xu, Ms. Preetika D. Banerjee and Ms. Surabhi Chaturvedi (WHO SEARO) for their valuable and remarkable contribution in the survey. We would also like to thank all NHRC staff who helped during conduction of this study.

## Author Contributions

**Conceptualization:** Bihungum Bista, Meghnath Dhimal, Saroj Bhattarai, Tamanna Neupane, Achyut Raj Pandey, Anjani Kumar Jha.

**Data curation:** Nick Townsend.

**Formal analysis:** Bihungum Bista, Meghnath Dhimal, Saroj Bhattarai, Yvonne Yiru Xu, Achyut Raj Pandey.

**Funding acquisition:** Anjani Kumar Jha.

**Investigation:** Meghnath Dhimal, Pradip Gyanwali, Anjani Kumar Jha.

**Methodology:** Pradip Gyanwali, Anjani Kumar Jha.

**Project administration:** Pradip Gyanwali, Anjani Kumar Jha.

**Supervision:** Pradip Gyanwali, Anjani Kumar Jha.

**Validation:** Nick Townsend.

**Visualization:** Tamanna Neupane.

**Writing – original draft:** Bihungum Bista, Meghnath Dhimal, Saroj Bhattarai, Tamanna Neupane, Yvonne Yiru Xu.

**Writing – review & editing:** Bihungum Bista, Meghnath Dhimal, Saroj Bhattarai, Tamanna Neupane, Yvonne Yiru Xu, Achyut Raj Pandey, Nick Townsend, Pradip Gyanwali, Anjani Kumar Jha.

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
