## [Decision Letter · Decision Letter 0]

6 Apr 2021

PONE-D-21-06886

Prevalence of Noncommunicable Diseases Risk Factors and their Determinants: Results from STEPS survey 2019, Nepal

PLOS ONE

Dear Dr. Dhimal,

Thank you for submitting your manuscript to PLOS ONE. After careful consideration, we feel that it has merit but does not fully meet PLOS ONE’s publication criteria as it currently stands. Therefore, we invite you to submit a revised version of the manuscript that addresses the points raised during the review process.

A **rebuttal letter** that responds to **EACH** point raised by the academic editor and reviewer(s). You should upload this letter as a separate file labeled 'Response to Reviewers'.A **marked-up copy** of your manuscript that highlights changes made to the original version. You should upload this as a separate file labeled 'Revised Manuscript with Track Changes'.An **unmarked version** of your revised paper without tracked changes. You should upload this as a separate file labeled 'Manuscript'.

We look forward to receiving your revised manuscript.

Kind regards,

Brecht Devleesschauwer

Academic Editor

PLOS ONE

Additional Editor Comments:

In addition to the reviewer's comments, please also consider the following:

1. The write-up of the manuscript is quite sloppy, with several typos and grammatical and punctuation errors. We therefore suggest you thoroughly copy-edit your manuscript for language usage, spelling, and grammar. If you do not know anyone who can help you do this, you may wish to consider employing a professional scientific editing service.

- A copy of your manuscript showing your changes by either highlighting them or using track changes

- A clean copy of the edited manuscript

2. Why was Poisson regression used to analyze the binary data, instead of logistic regression? Using Poisson regression for binary data underestimates variance, and hence leads to higher type 1 error.

In your revision note, please include EACH of the reviewer and editor comments, provide your reply, and when relevant, include the modified/new text (or motivate why you decided not to modify the text). Note that failure to do so may result in a rejection of the manuscript.

Journal Requirements:

3. We note you have included a table to which you do not refer in the text of your manuscript. Please ensure that you refer to Table 2 and 4 in your text; if accepted, production will need this reference to link the reader to the Table.

Reviewers' comments:

Reviewer's Responses to Questions

**Comments to the Author**

1. Is the manuscript technically sound, and do the data support the conclusions?

Reviewer #1: Yes

2. Has the statistical analysis been performed appropriately and rigorously? 

Reviewer #1: Yes

3. Have the authors made all data underlying the findings in their manuscript fully available?

Reviewer #1: Yes

4. Is the manuscript presented in an intelligible fashion and written in standard English?

Reviewer #1: Yes

5. Review Comments to the Author

Reviewer #1: Introduction

The rationale for reporting this study could be reframed better.

For ex. There is no link between the multi sectoral plan and current STEPS survey; linking these two might give better clarity to understand the need for present STEPS survey

Methods

Sampling methods adapted in previous STEPS surveys are different from the current survey. The change in context should get mentioned for the readers to compare and interpret these two surveys

As per methods details provided in the reference, the sample size was estimated to cover each province with equal number of participants (around 900 each). If it is so, why there are 20% participants from some province and 8% from some province. Please clarify.

As province and ecological belts were considered as independent variables across all NCD risk factors give a brief write up regarding the characteristics of these provinces and ecological belts which are relevant to understand the burden of NCD risk factors.

Though sample size and sampling methods were published elsewhere it is better to give a brief description on the sample size and selection of PSUs.

Under Ethical concern, for minors it require assent from the adolescents and consent from parents. Hence it needs to be clarified accordingly. Avoid the tern assent consent.

Results:

In this survey 15-29 years age group is predominant which is entirely different from the previous STEPS survey or Nepal population distribution based on age.

Please clarify how this over representation from this age group has happened.

Two decimals in results make the tables too crowded. It does not become reader friendly.

There is no display of gender disaggregated results within the province. But the discussion section presented in those lines.

Tobacco use and smokers are used inter changeably. Please use the term consistently

For fruits & vegetables intake, when everything is more than 97% sub group disaggregation does not help much (line no 221 to 223) could be avoided.

Table 2: As majority of the times the number in the sub group is going to remain same across all NCD risk factors it seems there is redundancy the way the n is represented. Instead of total number in the sub group it is preferred to give number with the particular risk factor present under the column n.

Discussion

Comparison of smoking with previous STEPS survey is incomplete.

The speculation for higher prevalence of smoking in Nepal is not justified well. (line No 268-274). Actually the prevalence in Nepal is more that should be explained by extent of lacunae in implementation of current tobacco legislative measures.

Adjusted prevalence ratio or adjusted relative risk has been calculated but it has been interpreted as if they were odds ratios. (line No 322-324, 334-335, 337,350, 353)

Introduction

The rationale for reporting this study could be reframed better.

For ex. There is no link between the multi sectoral plan and current STEPS survey; linking these two might give better clarity to understand the need for present STEPS survey

Methods

Sampling methods adapted in previous STEPS surveys are different from the current survey. The change in context should get mentioned for the readers to compare and interpret these two surveys

As per methods details provided in the reference, the sample size was estimated to cover each province with equal number of participants (around 900 each). If it is so, why there are 20% participants from some province and 8% from some province. Please clarify.

As province and ecological belts were considered as independent variables across all NCD risk factors give a brief write up regarding the characteristics of these provinces and ecological belts which are relevant to understand the burden of NCD risk factors.

Though sample size and sampling methods were published elsewhere it is better to give a brief description on the sample size and selection of PSUs.

Under Ethical concern, for minors it require assent from the adolescents and consent from parents. Hence it needs to be clarified accordingly. Avoid the tern assent consent.

Results:

In this survey 15-29 years age group is predominant which is entirely different from the previous STEPS survey or Nepal population distribution based on age.

Please clarify how this over representation from this age group has happened.

Two decimals in results make the tables too crowded. It does not become reader friendly.

There is no display of gender disaggregated results within the province. But the discussion section presented in those lines.

Tobacco use and smokers are used inter changeably. Please use the term consistently

For fruits & vegetables intake, when everything is more than 97% sub group disaggregation does not help much (line no 221 to 223) could be avoided.

Table 2: As majority of the times the number in the sub group is going to remain same across all NCD risk factors it seems there is redundancy the way the n is represented. Instead of total number in the sub group it is preferred to give number with the particular risk factor present under the column n.

Discussion

Comparison of smoking with previous STEPS survey is incomplete.

The speculation for higher prevalence of smoking in Nepal is not justified well. (line No 268-274). Actually the prevalence in Nepal is more that should be explained by extent of lacunae in implementation of current tobacco legislative measures.

Adjusted prevalence ratio or adjusted relative risk has been calculated but it has been interpreted as if they were odds ratios. (line No 322-324, 334-335, 337,350, 353)

6. PLOS authors have the option to publish the peer review history of their article (what does this mean?). If published, this will include your full peer review and any attached files.

Reviewer #1: No

---

## [Author Response · Author response to Decision Letter 0]

29 May 2021

28 May 2021 

Prof. Brecht Devleesschauwer

Academic Editor

PLOS ONE

Re: PONE-D-21-06886 (R1) Prevalence of Noncommunicable Diseases Risk Factors and their Determinants: Results from STEPS survey 2019, Nepal

Dear Professor Devleesschauwer

Thank you very much for your email of 7 April 2021 and comments on our manuscript. We have carefully revised the manuscript in response to the extensive and insightful comments we received from you and reviewer. 

In particular, the reviewer provided constructive comments with some corrections. In the revised manuscript version, all the suggested corrections are made. We have also revised the content of introduction, methodology, result and discussion section according to your and reviewer advice and have paid special attention to correcting all typological errors. Appended below is the list of all yours and reviewer comments along with our responses to each point.

We hope that this revised version will be suitable for publication in PLoS ONE. 

Yours sincerely,

Meghnath Dhimal, PhD 

 

Editors comments

1. The write-up of the manuscript is quite sloppy, with several typos and grammatical and punctuation errors. We therefore suggest you thoroughly copy-edit your manuscript for language usage, spelling, and grammar. If you do not know anyone who can help you do this, you may wish to consider employing a professional scientific editing service.

- A copy of your manuscript showing your changes by either highlighting them or using track changes

- A clean copy of the edited manuscript

**Thank you so much for your feedback and suggestions. We have carefully revised our manuscript and our manuscript is proof read by Professor Nick Townsend who is native speaker of English from United Kingdom. Based on his significant contribution in our manuscript, we have included him a co-author in our revised manuscript. Next as per request of WHO colleagues, we have transferred WHO affiliated co-authors in acknowledgements section. 

2. Why was Poisson regression used to analyze the binary data, instead of logistic regression? Using Poisson regression for binary data underestimates variance, and hence leads to higher type 1 error.

** Thank you for your feedback. Firstly,to establish the comparability with previous round survey findings (https://journals.plos.org/plosone/article?id=10.1371/journal.pone.0134834) which has also used Poisson regression has motivated us to use the Poisson regression in our analysis. Secondly, the Adjusted Prevalence Ratios rather than odds ratios allowed us to compare the relative strengths of association in a manner that was not biased by whether a risk factors was rare or common. Finally, as the outcomes gets frequent, there is chances of underestimates of variance, to combat with under estimates of variance due to frequent occurrence of events/outcomes, the robust variance estimates Huber's sandwich estimator has been used while running (robust) Poisson regression in STATA i.e., vce (robust). 

In your revision note, please include EACH of the reviewer and editor comments, provide your reply, and when relevant, include the modified/new text (or motivate why you decided not to modify the text). Note that failure to do so may result in a rejection of the manuscript.

Journal Requirements:

*Thank you so much for your suggestion and we have followed Journal’s author guidelines strictly. 

**Thank you so much for your comments and suggestion. The data of this study is available from the WHO NCD Microdata Repository (https://extranet.who.int/ncdsmicrodata/index.php/catalog/771/data_dictionary). 

3. We note you have included a table to which you do not refer in the text of your manuscript. Please ensure that you refer to Table 2 and 4 in your text; if accepted, production will need this reference to link the reader to the Table.

**Thank you so much for your suggestion and we have cited all tables in our revised manuscript. 

Reviewers Comment

Introduction

The rationale for reporting this study could be reframed better.

For ex. There is no link between the multi sectoral plan and current STEPS survey; linking these two might give better clarity to understand the need for present STEPS survey

** Thank you for your suggestion. We have added following text in our introduction section 

In order to track progress on prevention and control of NCD risk factors over the years, the multi-sectoral NCD action plan has included NCD STEPS survey to be conducted in every five years and as follow up of NCD STEPS survey 2013, this survey was conducted. 

Methods 

Sampling methods adapted in previous STEPS surveys are different from the current survey. The change in context should get mentioned for the readers to compare and interpret these two surveys 

** Thank you for your suggestion. Necessary changes has been made in the revised masncuript. 

Sampling for the survey took into consideration the current federal structure of Nepal, such that findings could be generalized to the provincial levels. The household listing operation was carried out in 259 PSUs, in order to develop a sampling frame for selection of individual households at the second stage. 

As per methods details provided in the reference, the sample size was estimated to cover each province with equal number of participants (around 900 each). If it is so, why there is 20% participants from some province and 8% from some province. Please clarify.

** All the percentage displayed in findings are adjusted using sample weighing and population weighting. So, province wise distribution of participants will be similar to like province population distribution. 

As province and ecological belts were considered as independent variables across all NCD risk factors give a brief write up regarding the characteristics of these provinces and ecological belts which are relevant to understand the burden of NCD risk factors. 

** Thank you for your suggestion. Necessary changes has been made in revised manuscript 

Study settings: Nepal is a landlocked country situated in Southern Asia between India and China. The country runs from a plain area in the South, known as Terai, to the mountainous area of the Himalayas in the North, with a hilly region in between the two. Administratively, Nepal is comprised of 7 provinces, 77 districts and 753 local bodies.

Though sample size and sampling methods were published elsewhere it is better to give a brief description on the sample size and selection of PSUs. 

** Thank you for your suggestion. Necessary changes have been made in method section of the revised manuscript.

A total of 259 wards were selected as the primary sampling units (PSU) at the first stage, maintaining 37 PSUs from every province. The household listing operation was carried out in 259 PSUs, in order to develop a sampling frame for selection of individual households at the second stage

Under Ethical concern, for minors it require assent from the adolescents and consent from parents. Hence it needs to be clarified accordingly. Avoid the tern assent consent.

** Thank you for your suggestion. Necessary changes has been made in the revised manuscript 

Ethical approval to conduct this survey was granted from the Ethical Review Board (ERB) of the Nepal Health Research Council (NHRC), Government of Nepal (Registration number 293/2018). Written informed consent was obtained from each participant before they enrolled in the survey. In case of minors (under 18 years old) both assent from the research participants and consent from their parents (legal guardian) was obtained, as per national ethical guidelines for health research in Nepal. We also took administrative approval from federal, provincial and local governments, as per the need. The confidentiality of all information gathered was maintained. Any waste generated during the laboratory procedures was properly disinfected using aseptic techniques before being safely disposed of. All blood and urine samples were discarded after completing biochemical measurements.

Results:

In this survey 15-29 years age group is predominant which is entirely different from the previous STEPS survey or Nepal population distribution based on age.

Please clarify how this over representation from this age group has happened.

** There were some error in reporting the percentage, it has been corrected in the revised manuscript. Regarding percentage of 15-29 years age group, it is a weighted percentage i.e 44.9% which is quite near to previous round of survey i.e 46.5% (https://journals.plos.org/plosone/article?id=10.1371/journal.pone.0134834)

Two decimals in results make the tables too crowded. It does not become reader friendly. 

There is no display of gender disaggregated results within the province. But the discussion section presented in those lines. 

** Thank you. It has been corrected in the revised manuscript (especially on table 2). Regarding gender disaggregation within the province, that was a mistake in reporting in discussion. It has been revised. 

We found that participants residing in Province 1, Province 2, Bagmati province, and Lumbini province were less likely to smoke than those in Sudurpaschim province, with this finding aligning with another national level survey 

Tobacco use and smokers are used inter changeably. Please use the term consistently

** Thank you for your comments. It has been corrected in revised version of manuscript with a word smoker only.

For fruits & vegetables intake, when everything is more than 97% sub group disaggregation does not help much (line no 221 to 223) could be avoided. 

** Thank you for your suggestion. Since every variable has been described individually as well as to maintain comparability with last round survey findings, we think it is wise to describe briefly.

Table 2: As majority of the times the number in the sub group is going to remain same across all NCD risk factors it seems there is redundancy the way the n is represented. Instead of total number in the sub group it is preferred to give number with the particular risk factor present under the column n. 

** Thank you for your suggestion. As for every variable sample size (n) has varied due to missing of data, so, it has been reported separately for every variable.

Discussion

Comparison of smoking with previous STEPS survey is incomplete. 

The speculation for higher prevalence of smoking in Nepal is not justified well. (line No 268-274). Actually, the prevalence in Nepal is more that should be explained by extent of lacunae in implementation of current tobacco legislative measures. 

** Thank you for your suggestions. Necessary changes has been made in the revised manuscript. 

Adjusted prevalence ratio or adjusted relative risk has been calculated but it has been interpreted as if they were odds ratios. (line No 322-324, 334-335, 337,350, 353).

** Thank you for your valuable suggestions. In the above-mentioned line number as well as in other line number, it has been corrected in the revised manuscript. 

This study has the lot of scope to bring out the change or impact of previous strategic plan through comparison of previous STEPs survey with the current. In similar lines, if there is reduction or increase in the NCD risk factor level the reasons could have been explained better.

** Thank you for your valuable suggestions. As per suggestions, necessary amendment has been made in the revised manuscript with possible explanation wherever possible. But in case of some variables, comparison is not possible due to difference in methodological or instrumental difference (eg. Level of Cholesterol), that has been explained wherever necessary as a limitation of the study.

---

## [Decision Letter · Decision Letter 1]

9 Jun 2021

Prevalence of noncommunicable diseases risk factors and their determinants: Results from STEPS survey 2019, Nepal

PONE-D-21-06886R1

Dear Dr. Dhimal,

We’re pleased to inform you that your manuscript has been judged scientifically suitable for publication and will be formally accepted for publication once it meets all outstanding technical requirements.

Kind regards,

Brecht Devleesschauwer

Academic Editor

PLOS ONE

Additional Editor Comments (optional):

Reviewers' comments:

Reviewer's Responses to Questions

**Comments to the Author**

1. If the authors have adequately addressed your comments raised in a previous round of review and you feel that this manuscript is now acceptable for publication, you may indicate that here to bypass the “Comments to the Author” section, enter your conflict of interest statement in the “Confidential to Editor” section, and submit your "Accept" recommendation.

Reviewer #1: All comments have been addressed

2. Is the manuscript technically sound, and do the data support the conclusions?

Reviewer #1: Yes

3. Has the statistical analysis been performed appropriately and rigorously? 

Reviewer #1: Yes

4. Have the authors made all data underlying the findings in their manuscript fully available?

Reviewer #1: Yes

5. Is the manuscript presented in an intelligible fashion and written in standard English?

Reviewer #1: Yes

6. Review Comments to the Author

Reviewer #1: The manuscript has been substantially revised in terms of language and Reviewers comments.

The survey methods and rationale section has been revised according to previous comments. Similarly, the findings are appropriately interpreted as adjusted Prevalence ratio

7. PLOS authors have the option to publish the peer review history of their article (what does this mean?). If published, this will include your full peer review and any attached files.

Reviewer #1: No

---

## [Editor Report · Acceptance letter]

22 Jul 2021

PONE-D-21-06886R1 

Prevalence of non-communicable diseases risk factors and their determinants: Results from STEPS survey 2019, Nepal 

Dear Dr. Dhimal:

I'm pleased to inform you that your manuscript has been deemed suitable for publication in PLOS ONE. Congratulations! Your manuscript is now with our production department. 

Kind regards, 

on behalf of

Prof. Dr. Brecht Devleesschauwer 

Academic Editor

PLOS ONE